# T-cell Redirecting Therapies for the Treatment of B-cell Lymphomas: Recent Advances

**DOI:** 10.3390/cancers13174274

**Published:** 2021-08-25

**Authors:** Ondine Messéant, Roch Houot, Guillaume Manson

**Affiliations:** Department of Hematology, University Hospital of Rennes, 35000 Rennes, France; ondine.messeant@chu-rennes.fr (O.M.); Roch.HOUOT@chu-rennes.fr (R.H.)

**Keywords:** lymphoma, immunotherapy, CAR T-cells, bispecific antibody, T-cells engager

## Abstract

**Simple Summary:**

B-cell non-Hodgkin lymphomas (NHL) include many diseases with distincts pathogenic mechanisms, prognoses and management. Most patients benefit generally from efficient therapies allowing cure or prolonged remission. However, when they are refractory or relapse after standard therapy, they harbor a poor prognosis. In last decades, numerous novel immunotherapies have been developed with the aim of redirecting T-cell specificity against tumor antigens. Latest data on CAR T-cells confirm their efficacy and their safety in this setting. In addition, trials with bispecific antibodies are also ongoing for these patients, with encouraging premiminary findings, whether before or after CAR T-cells treatment. Here, we review the main results of CAR T-cells and bispecific T-cell engagers studies in the B-cell non-Hodgkin lymphomas setting. These advances in immunotherapies have transformed diffuse large B-cell lymphomas prognosis and will process indolent NHL’s future. Results with such treatments could lead to a new standard of care for those patients who are often heavily pretreated.

**Abstract:**

T-cell specificity can be redirected against tumor antigens either ex vivo using engineered chimeric antigen receptor (CAR) T-cells or in vivo by bridging natural T-cells and tumor cells with bispecific T-cell engager (TCE) antibodies. Currently, four CAR T-cells have been approved by the FDA for the treatment of B-cell lymphomas, including diffuse large B cell lymphomas (DLBCL), mantle cell lymphoma (MCL), and follicular lymphoma (FL). No TCE have yet been approved for the treatment of B-cell lymphomas. However, at least four of them are in clinical development and show promising activity. Here, we review the most recent advances of CAR T-cells and TCE in the treatment of B-cell lymphomas.

## 1. Introduction

B-cell non-Hodgkin lymphomas (B-NHL) benefit from efficient therapies allowing cure or prolonged remission in most patients. However, patients who are refractory or relapse (R/R) after standard therapies harbor a poor prognosis, notably patients with R/R diffuse large B-cell lymphoma (DLBCL). Indeed, the SCHOLAR-1 study reported by Crump et al. found a median overall survival from salvage therapy initiation of 6.3 months for patients with R/R DLBCL [1]. On the other hand, patients with indolent lymphoma cannot be cured in most cases and may need multiple lines of treatment, with shorter periods of remission as time goes on.

New therapeutic approaches are therefore needed to improve the prognosis of these patients. In recent years, novel therapies have been developed to mobilize and re-direct the immune system against malignant B-cells in patients who fail standard therapies. These therapies can be achieved with chimeric antigen receptor T-cells (CAR-T) and bispecific T-cell engager (TCE) antibodies. Currently, four CAR T-cells—axicabtagene ciloleucel (Axi-cel, Yescarta^®^, Hoofddorp, The Netherlands), tisagenlecleucel (Tisa-cel, Kymriah^®^, Basel, Switzerland), lisocabtagene maraleucel (Liso-cel, Breyanzi^®^, Seattle, WA, US), and brexucabtagene autoleucel (Brexu-cel, Tecartus^®^, Hoofddorp, The Netherlands)—have been approved by the FDA for the treatment of B-cell lymphomas, including diffuse large B-cell lymphomas (DLBCL), mantle cell lymphoma (MCL), and follicular lymphoma (FL) (Table 1). No TCE have yet been approved for the treatment of B-cell lymphomas. 

Here, we review the most recent advances of CAR T-cells and TCE in the treatment of B-cell lymphomas. These advances in immunotherapies have transformed DLBCL prognosis and process indolent NHLs in the future. Results with such therapies may lead to a new standard of care for heavily pretreated patients

## 2. CAR T-cells

### 2.1. Current CAR T-cells in DLBCL

#### 2.1.1. Confirmed Safety and Efficacy in DLBCL

Patients with refractory or relapsed DLBCL have a poor life expectancy, especially when they present refractory or relapsed (R/R) disease after salvage treatment. The international retrospective SCHOLAR-1 study analyzed data from 636 patients with refractory DLBCL to frontline therapy [1]. In this population, the overall response rate (ORR) to a new line of therapy was 26%, including 7% of complete responses (CR). Only 20% of patients were alive at 2 years, with a median overall survival (OS) of 6.3 months.

Based on the pivotal ZUMA-1 and JULIET trials, two CD19 CAR T-cells—axi-cel (Yescarta^®^) and tisa-cel (Kymriah^®^)—were approved in 2017 and 2018, respectively, for the treatment of DLBCL patients who failed at least two lines of therapy [2,3] (Table 1). 

These studies were recently updated. After a median follow-up of 51.1 months for ZUMA-1 and 40.3 months for JULIET, the overall response (OR) and complete response (CR) rates were 83% and 58% for axi-cel, and 54% and 40% for tisa-cel, respectively. The overall survival (OS) was 44% at 4 years for axi-cel and 36% at 3 years for tisa-cel [4,5] (Table 2). These results confirm the long-term efficacy of CAR T-cells in DLBCL, with few relapses beyond one year. These remissions persisted even though most patients in CR had recovered normal B-cells 9 and 12 months after axi-cel and tisa-cel infusion, respectively [4]. 

Regarding adverse effects, long-term safety analysis of these CAR T-cells revealed no unexpected toxicities. 

The main early toxicities relative to CAR T-cells treatment are inflammatory toxicities, such as cytokine release syndrome (CRS), immune effector cell-associated neurotoxicity (ICANS), and cytopenias [7,8]. Following infusion, CAR T-cells recognize the target that activates them and induces the secretion of various inflammatory factors (GM-CSF, IL-6, IL-1b, C-reactive Protein [7,8]. CRS of grade ≥ 3 was observed in 11% of patients with Yescarta^®^ and 23% with Kymriah^®^. Grade ≥ 3 ICANS was present in 32% and 12% of each respective treatment [4,5] (Table 2).

The management of these toxicities depends on the intensity of the symptoms and consists of anti-inflammatory and symptomatic treatments [7]. Currently, tocilizumab, an IL-6 receptor antagonist, is the only FDA-approved therapy for CRS treatment. Corticosteroids are also prescribed in cases where tocilizumab is insufficient to contain symptoms of severe CRS. Furthermore, other anti-inflammatory molecules have been studied, such as siltuximab, an IL-6 antagonist, or anakinra, an antibody-based IL-1 receptor (IL-1R) antagonist that can cross the blood–brain barrier (BBB) [7,8]. The occurrence of ICANS may be simultaneous with or independent of CRS. Dexamethasone is usually used to reverse ICANS because it appears to penetrate the BBB [8].

Long-term toxicity of CAR-T cells therapy remains uncommon and manageable: In the ZUMA-1 study, among the 101 patients treated with axi-cel, only one case of myelodysplasia and four cases of secondary cancer (unspecified) were reported in 2 years. In the 4-year analysis, no new cancer was reported. The main long-term concern was the risk of infection caused by B-cell aplasia and hypogammaglobulinemia secondary to CAR-T cells. Thirty percent of patients received, which are currently recommended for patients with recurrent infections [9]. No case of delayed CRS or delayed ICANS was described.

Furthermore, these results were confirmed by real-life studies in large cohorts of patients. Efficacy and safety were similar to those reported in ZUMA-1 (Table 3) and JULIET (Table 4) [10,11].

Two subgroup analyses, one from the ZUMA-1 study and another from the CIBMTR registry, demonstrated that CAR-T cell therapy was feasible in elderly patients, with similar efficacy and safety compared to younger patients [5,12,13]. 

#### 2.1.2. An Earlier Use of CAR T-cells in Therapeutic Strategies for DLBCL

Many studies have evaluated CAR T-cells in heavily pre-treated patients. Providing CAR T-cells earlier may allow the production of CAR T-cells from T-cells that have received limited chemotherapies, which may have enhanced functionality following leukapheresis. Earlier use also allows the treatment of patients with better performance status and lower tumor burden [14,15]. Three ongoing phase III studies compare CAR T-cells to autologous stem cell transplantation (auto-HSCT) in patients with DLBCL at first relapse: ZUMA-7 (axi-cel), BELINDA (tisa-cel), and TRANSFORM (liso-cel). Two recent press releases announced that two of these—ZUMA-7 and TRANSFORM—met their primary endpoint [16,17]. In addition, the ongoing TRANSCEND-PILOT-017006 and ALYCANTE trials evaluate liso-cel and axi-cel, respectively, as second-line therapies in DLBCL patients who are not eligible for auto-HSCT (NCT03483103 and NCT04531046).

Recently, the ZUMA-12 study evaluated axi-cel as part of the first-line treatment in patients with “high-risk” DLBCL. This population was defined by a double or triple hit, an IPI ≥ 3, and a positive interim PET scan (defined by a Deauville scale of 4 or 5) after two cycles of R-CHOP or an R-CHOP-like regimen [18]. Among the 32 infused patients, 53% had double or triple hit lymphoma, and 72% had an IPI score ≥ 3. The median age was 61 years. The ORR was 85%, including 74% CR; by comparison, the CR rate was 58% in ZUMA-1 [5]. After a median follow-up of 9.3 months, the median progression-free survival (PFS) and duration of response (DOR) were not reached. Toxicity appeared similar to the one reported in ZUMA-1. All patients experienced CRS, with 9% grade ≥ 3, and half of all patients received tocilizumab. Sixty-nine percent of the cohort experienced ICANS, including 25% grade ≥ 3. One-third of the patients received corticosteroids [5,19]. Locke et al reported that the presence of CCR7 + CD45RA + T-cells in the product correlated with efficacy in the ZUMA-1 trial [20]. Interestingly, the number of CCR7 + CD45RA + T-cells was approximately two and half times higher in ZUMA-12 compared with the ZUMA-1 trial, with greater peak expansion.

### 2.2. Novel Indications in B-cell Lymphomas

Outcomes following treatments with axi-cel and tisa-cel have led to developing other kinds of CAR T-cells with the aim of less toxicity. Thus, two new anti-CD19 CAR T-cells have emerged and recently been approved by the FDA.

Results of pivotal studies have also led to the extension of CAR T-cells to other B-cell lymphomas, such as mantle cell and indolent lymphomas.

#### 2.2.1. Diffuse Large B-cell Lymphoma

In February 2021, the FDA approved a third CAR T-cell, liso-cel (lisocabtagene-maraleucel, Breyanzi^®^) for the treatment of R/R DLBCL after at least two lines of systemic therapy (Table 1). Liso-cel, a CD19 CAR T-cell with a 4-1BB costimulatory domain, comprises an equal number of CD4 + and CD8 + T-cells. 

This approval is based on the results of Abramson’s pivotal study on TRANSCEND NHL-001 [6]. In this study, 269 patients were treated with liso-cel, representing the largest CAR T-cell trial to date. Eligibility criteria were more flexible than in the ZUMA-1 and JULIET studies: 19% had moderate renal failure, and 5% had moderate heart failure. While neuro-meningeal involvement was an exclusion criterion in ZUMA-1 and JULIET, six patients had central nervous system (CNS) involvement (allowed) in TRANSCEND NHL-001. More than 95% of patients were heavily pre-treated, with a median of more than five prior lines. The ORR was 73%, including 53% CR. The safety was excellent, with only 2% grade ≥ 3 CRS and 10% grade ≥ 3 ICANS (Table 2). Treatment-related deaths were reported in 3% of cases (one case of diffuse intra-alveolar hemorrhage, one case of pulmonary hemorrhage, one case of multivisceral failure, one case of cardiomyopathy, one case of progressive multifocal leukoencephalopathy, one case of septic shock, and one case of leukoencephalopathy). Given its limited toxicity, liso-cel has been evaluated in the outpatient setting. Godwin et al. reported the outcome of 22 patients treated with liso-cel as outpatients in the OUTREACH study [21]. Hospitalization was required for 50% of the outpatients. The median time to hospitalization was 5 days, and the duration of hospitalization was approximately 6 days.

#### 2.2.2. Mantle Cell Lymphoma

The phase II ZUMA-2 trial evaluated brexu-cel (KTE-X19, TECARTUS^©^) in R/R MCL [22,23]. Brexu-cell is a CD19 CAR-T with a construct similar to axi-cel but with a manufacturing process that includes a T-cell selection and lymphocyte enrichment to exclude circulating malignant B-cells. 

Sixty-eight patients received brexu-cel infusion at a dose of 2 × 10^6^/kg after lymphodepleting chemotherapy. Most patients had poor prognosis features, including stage IV disease (85%), intermediate or high-risk MIPI (56%), blastoid MCL (59%), high Ki67 (69%), or TP53 mutation (17%). Patients received at least one prior therapy, including an anthracycline or bendamustine-containing chemotherapy regimen, an anti-CD20 monoclonal antibody, and a BTK inhibitor (BTKi). After a median follow-up of 17.5 months, the ORR was 92%, including 67% of CR. PFS and OS were 59% and 76% at 15 months, respectively. After 2 years of follow-up, 48% of the patients remained in remission. CRS occurred in 91% of the patients, including 15% grade ≥ 3, and ICANS occurred in 63%, including 31% grade ≥ 3 (Appendix A). Based on these results, brexu-cel was approved by the FDA in July 2020 (Table 1).

In the TRANSCEND NHL-001 trial’s cohort mentioned above, 32 MCL patients were treated with liso-cel after at least two lines of systemic therapy, including a BTKi, an alkylating agent, and an anti-CD20 antibody [24]. Thirty-one percent were refractory to ibrutinib and 25% to venetoclax. Thirty-one percent and 9% received prior auto- and allo-HSCT, respectively. In this population, a large proportion of MCL was considered high-risk: 41% were blastoid, almost three-quarters had a high Ki67, and 22% had a TP53 mutation. Secondary CNS involvement occurred in 30% of patients. Patients received liso-cel at a dose of 50 to 100 × 10^6^, resulting in an ORR of 84% and 66% of CR (Appendix A). There was no significant difference in response to the high-risk group. After a median follow-up of 5.9 months, the median DOR had not been reached. CRS occurred in 50% of patients, including 3% grade ≥ 3, and ICANS in 34%, including 12% grade ≥ 3. Liso-cel has not yet been approved for the treatment of MCL.

#### 2.2.3. Follicular Lymphoma

The phase II ZUMA-5 trial evaluated axi-cel in 146 patients with indolent lymphoma: 124 follicular lymphomas (FL) and 22 marginal zone lymphomas (MZL) [25]. Many of these patients had poor prognostic factors: 44% had a FLIPI between 3 and 5, 55% had disease progression within 24 months of the first treatment (POD24), and more than half had a high tumor burden (as defined by GELF criteria). Patients received axi-cel after at least two lines of therapy, including an anti-CD20 monoclonal antibody and an alkylating agent. The ORR was 92% with 76% CR. After a median follow-up of 17.5 months, OS was 92.9% at 12 months. CRS occurred in 82% of patients, including 7% grade ≥ 3. ICANS occurred in 60% of patients, including 19% grade ≥ 3 (Appendix A). These data led to the FDA approval of axi-cel for R/R FL in March 2021 (Table 1). Interestingly, 11 patients (9 FL and 2 MZL) who responded after the first infusion of axi-cel received a second infusion at relapse. Of note, all tumors retained CD19 expression at relapse. After the second infusion, the ORR was 100%, including 82% CR [26].

The ELARA study evaluated the efficacy of tisa-cel in R/R FL [27]. The study included 97 patients: 60% had a FLIPI ≥ 3, and 60% had a POD24. The ORR was 86.2%, including 66% CR [28]. The PFS was 76% at 6 months (Appendix A). There was no grade 3–4 CRS and only 1% grade 3–4 ICANS.

### 2.3. Novel CARs

#### 2.3.1. Bispecific CAR T-cells

Approximately a third of DLBCL relapses exhibit CD19 loss or down-regulation after CAR T-cell therapy [29]. Bispecific CAR T-cells have been developed to prevent or reduce the risk of immune escape by loss of target antigen. Shah et al. reported the results of a phase 1 trial evaluating bispecific CAR T-cells against CD19 and CD20, containing 4-1BB costimulatory domains [30]. Twenty-two adults with R/R B-NHL (50% DLBCL, 9% Richter’s transformation, 32% MCL, 14% CLL, and 4% FL) received the bispecific CAR T-cells at a dose of 2.5 × 10^5^ to 2.5 × 10^6^ cells/kg. This last dose was chosen for the expansion phase. Biopsies obtained from relapses in patients after anti-CD19 CAR T-cells therapy (but before bispecific CAR T-cells) revealed that approximately 30% of patients were CD19 negative. Safety was efficient with only 5% (one patient) grade 3–4 CRS and 14% (three patients) grade 3–4 ICANS. The ORR was 82% at day 28 post-infusion, including 64% CR. 

The ALEXANDER study evaluated AUTO3, bispecific CAR T-cells with two different CAR (bicistronic construct): one against CD19 with an OX40 costimulatory molecule and another against CD22, with a 4-1BB costimulatory molecule [31]. This trial evaluated the safety and efficacy of AUTO3 +/− pembrolizumab in patients with R/R DLBCL. Thirty-three patients received AUTO3 alone, AUTO3 plus three doses of pembrolizumab, or AUTO3 plus a single dose of pembrolizumab. Overall, CRS occurred in 11 (33%) patients and ICANS occurred in 3 (9%) patients (including one grade 3). No case of grade 3 or 4 CRS was described in the cohort. Also, no case of ICANS was observed in the group “AUTO3 plus pembrolizumab”. Among patients who received the highest dose of AUTO3 with pembrolizumab at day 1, the ORR and CR were 75% and 63%, respectively.

#### 2.3.2. Allogeneic CAR T-cells

Healthy donors may produce allogeneic CAR T-cells. Such CAR T-cells present theoretical advantages because they use healthy T-cells (not previously exposed to cancer nor chemotherapy) and are immediately available (“off-the-shelf”). The ALPHA study evaluated the safety and efficacy of ALLO-501, an allogeneic CAR-T. This treatment was deleted for TCR (to prevent GvH) and CD52 to allow selective lymphodepletion and prevent CAR-T cell rejection [32]. Patients received a conditioning regimen with fludarabine, cyclophosphamide, and ALLO-647, an anti-CD52 depleting antibody. A total of 22 DLBCL and FL patients were infused with ALLO-501. The ORR obtained was 63%, including 57% CR. The safety was efficient with only 5% grade ≥ 3 CRS and no case of ICANS. No graft versus host disease (GvHD) was reported. However, the median follow-up was short (3.8 months).

PBCAR0191 is another allogeneic CAR T-cell recently reported by Shah et al. [33]. This treatment was evaluated in 17 patients with R/R CD19-positive NHL, including 85% aggressive NHL. Eleven of the 17 treated patients received a « standard lymphodepletion » (sLD) with fludarabine 30 mg/m^2^/d for 3 days + cyclophosphamide 500 mg/m^2^/d for 3 days, while the six remaining patients received an « enhanced lymphodepletion» (eLD) with fludarabine 30 mg/m^2^/d for 4 days + cyclophosphamide 1000 mg/m^2^/d for 3 days. Although the number of patients remains limited, the initial results suggested that a more intense lymphodepletion provides a better response rate (ORR = 89% vs. 50% with eLD and sLD, respectively) and a better CAR-T cell expansion. Toxicity was increased in the eLD vs. sLD subgroup with 45% vs. 50% CRS, 18% vs. 33% ICANS, and 18% vs. 0% grade 3 infections, respectively. There were no cases of GvHD in either group.

#### 2.3.3. CAR NK-cells

NK cells can also be engineered to express a CAR. Since they are deprived of TCR, they are not expected to induce graft-versus-host disease, and as such, can be produced from allogeneic donors. Liu et al. generated CD19 allogeneic NK-cells from cord blood transduced to secrete interleukin-15, enhancing their in vivo expansion and persistence [34]. These CAR NK-cells can recognize and eliminate cancer cells through their CAR and/or innate receptors, reducing the risk of tumor escape [35]. In this phase I/II study, 11 patients with CD19 + lymphoid malignancies (five CLL including two Richter’s syndromes and six DLBCL) received CAR-NK cells with HLA mismatch [34]. CAR-NK cells were infused at a dose of 1 × 10^5^, 1 × 10^6^, or 1 × 10^7^ cells/kg. With a median follow-up of 13.8 months, the ORR rate was 73%, including 64% CR. Responses occurred within 30 days of CAR-NK infusion, regardless of the dose. Surprisingly, despite the HLA mismatch, circulating CAR-NK cells could still be detected in the blood up to 12 months after infusion. Furthermore, the safety was effective, with no CRS, ICANS, nor GvHD reported.

## 3. T-cell Engagers

Besides CAR T-cells, other immunotherapies such as T-cell engagers have emerged in the last decades. Bispecific T-cell engager (TCE) antibodies can recruit and activate T-cells (usually through CD3) at the tumor site (usually through CD19 or CD20). Blinatumomab, a CD3/CD19 T-cell engager, was initially developed and approved for B-ALL treatment and demonstrated some efficacy in B-NHL, with an ORR ranging from 37% to 69% and some long term remissions [36,37,38,39]. However, the doses required in B-NHL are much higher (approximately four times) than in B-ALL, resulting in significant ICANS (≥20% grade 3–4). In addition, its scFv format (deprived of Fc) results in a very short half-life—approximately 2 h—and requires continuous IV infusion. To address these issues, novel bispecific antibodies have been developed containing an Fc-fragment directed against CD20, which is thought to reduce neurotoxicity. The main characteristics of these novel TCEs are summarized in Table 5.

### 3.1. Mosunetuzumab

Mosunetuzumab (RG7828) is a fully humanized CD20/CD3 TCE evaluated in patients with R/R DLBCL, including patients who failed a CAR T-cell therapy (Table 5). In this phase I/Ib trial (GO29781), 270 patients with R/R NHL were included [40]. CRS was reported in 29% of patients, and ICANS in 4%. Among patients with indolent NHL, the ORR was 62.7%, including 43.3% CR. Most of these responses (83%) were maintained at 26 months. In aggressive NHL, ORR was 37.1%, including 19.4% CR. Seventy-one percent remained in CR at 16 months. In the subset of patients who relapsed after CAR T-cells, the ORR was 39%, including 22% CR. In these patients, immune monitoring showed an increase in CAR T-cells after mosunetuzumab infusion.

Mosunetuzumab has also been evaluated as a first-line therapy in DLBCL patients ineligible for optimal chemotherapy [44]. Patients had to be over 80 or 60 years old with impaired activity of daily living (ADL), instrumental ADL (iADL), cardiac, renal, or hepatic function, precluding the use of full-dose intensive chemotherapy (CIT). They received mosunetuzumab intravenously in escalating doses at D1, D8, and D15 on the first cycle, followed by a fixed antibody dose on each 21-day cycle. A pre-phase of treatment with corticosteroids and/or vincristine was allowed. Nineteen patients were evaluable, and the median age was 84 years. The most frequent AEs were CRS (47%, all grade 1) and rash (21%). Grade 2 ICANS was reported in 5% of the patients and associated with CRS. No grade ≥ 3 ICANS nor grade 5 AEs were reported. The ORR was 58%, including 42% CR. Forty-two percent of patients had to discontinue mosunetuzumab prematurely due to progressive disease (PD) between cycles 2 and 6.

### 3.2. Glofitamab

Glofitamab (RG6026) is a CD3/CD20 TCE with a 2:1 format: a bivalent binding to CD20 on B-cells, and a monovalent binding to CD3 on T-cells [45]. A phase I/Ib study recently evaluated the safety, pharmacokinetics, and maximum tolerated dose of glofitamab (Table 5) [41]. The study cohort included 171 adults with R/R B-NHL: 127 aggressive B-NHL (DLBCL, TFL, or other aggressive histology) and 44 indolent NHL. Patients had been heavily pre-treated, with a median of three prior lines. The majority were refractory to their last treatment and prior anti-CD20 therapy. Patients were pre-treated with one dose of obinutuzumab 7 days before the first infusion of glofitamab (0.005 to 30 mg) to reduce the risk of severe CRS. Glofitamab was administered intravenously in increasing doses during the first cycle (D1 and D8) and then at the target dose from the second cycle onwards (2.5/10/16 mg or 2.5/10/30 mg). Patients then received an infusion every three weeks for up to 12 cycles. The most frequent AEs were CRS, fever, cytopenias (neutropenia and thrombocytopenia), and hypophosphatemia. Grade 3 or 4 CRS occurred in 3.5% of patients and grade 3 ICANS in 1.2%. Among patients who received the dose chosen for phase II, the ORR was 65.7%, including 57.1% CR. This response was maintained in 84.1% of patients after a median follow-up of 27 months [41].

### 3.3. Odronextamab

Odronextamab (REGN1979) is a fully human CD20/CD3 TCE that was tested in a phase I study (Table 5) [42]. This treatment was administered intravenously in 127 patients with B-NHL (56% of DLBCL) at doses ranging from 0.03 to 320 mg (with escalating doses at weeks 1 and 2, then fixed doses). Patients relapsed or were refractory after at least one line of therapy (median of three prior lines). Eighty percent were refractory to their last treatment and almost a quarter (22.8%) were treated with CAR T-cells. No dose-limiting toxicities were observed, and the maximum tolerated dose was not reached. The main AEs were fever (76.4%), CRS (62.2%, including 6.3% grade 3, and 0.8% grade 4) and chills (48%). Grade 3 ICANS occurred in 2.3% of cases. In patients with R/R FL (*n* = 28) who received doses ≥5 mg, the ORR with a median follow-up of 3.9 months was 92.9%, including 75% CR. The median duration of complete response was 8.1 months. For patients with R/R DLBCL receiving a odronextamab dose of at least ≥80 mg, the CR rate was 60% for those who had not received prior CAR-T therapy (with a median duration of CR of 9.5 months) and 23.8% for those who had received prior CAR T-cells (*n* = 21).

### 3.4. Epcoritamab

Epcoritamab is a fully human CD20/CD3 TCE administered subcutaneously (SC) (Table 5) [43,46]. The SC route is more convenient and expected to reduce severe CRS risk by inducing a more gradual increase of antibody in plasma cytokine levels and lower peak levels. Clausen et al. reported a phase I/II trial evaluating epcoritamab in 68 patients with R/R CD20-positive B-NHL (67.6% DLBCL, 17.6% FL, and 5.9% MCL) [43]. All subjects were refractory to their last anti-CD20 antibody therapy, and 9% received prior CAR T-cells therapy. Patients received one fixed-dose SC injection of epcoritamab per cycle every 28 days, until disease progression or unacceptable toxicity. The most common treatment-emergent AEs were fever (69%), injection site skin reactions (47%), and fatigue (44%). All CRS were grade 1 or 2 (59%). Transient ICANS was reported in 6% of patients (3% of grade 3). There was no dose-limiting toxicity nor treatment-related death. Based on these analyses, the dose selected for phase 2 was 48 mg. Among the 11 DLBCL patients who received epcoritamab at ≥48 mg, the ORR was 91%, including 55% CR. All patients who had been previously treated with CAR T-cells (*n* = 4) achieved a response (two CR and two PR). Among the 12 FL patients who received epcoritamab at ≥12 mg, the ORR was 80%, including 60% CR. Finally, two out of four MCL patients experienced an objective response (one CR and one PR).

## 4. Conclusions

CAR T-cells and bispecific TCE antibodies have emerged as new therapeutic modalities for the treatment of B-cell lymphomas. After initial approval, CAR T-cells confirmed their efficacy in DLBCL, suggesting further benefit when used earlier in the therapeutic strategy, and their indications extend to other B-cell lymphomas. Novel CARs, including new constructs (bispecific CAR T-cells) and different cell sources (allogeneic CAR T-cells, CAR NK-cells), are being developed, further optimizing these therapies. TCE antibodies can redirect T-cells in vivo against lymphoma B-cells. These “off-the-shelf” drugs are less advanced in their clinical development, but their initial results are promising. Undoubtedly, they will also be a part of the therapeutic armamentarium of B-cell lymphomas in the near future. Overall, the field of T-cell redirecting therapies is moving forward rapidly and should establish itself as a major treatment of B-cell lymphomas. 

## Figures and Tables

**Table 1 cancers-13-04274-t001:** CAR T-cells Currently approved by the FDA for the Treatment of B-cell Lymphoma.

CAR-T cells	Axi-cel (Yescarta^®^)	Tisa-cel (Kymriah^®^)	Brexu-cel (Tecartus^®^)	Liso-cel (Breyanzi^®^)
DLBCL	2017	2018	-	2021
MCL	-	-	2020	-
FL	2021	-	-	-

DLBCL, diffuse large B-cell lymphoma; FL, follicular lymphoma; MCL, mantle cell lymphoma.

**Table 2 cancers-13-04274-t002:** Updated CAR T-cells Pivotal Studies in DLBCL.

CAR-T cells		Axi-cel (Yescarta^®^)	Tisa-cel (Kymriah^®^)	Liso-cel (Breyanzi^®^)
Reference	Reference	Neelapu et al [2]	Scuster et al [3]	Abramson et al [6]
	Company	Kite-Gilead	Novartis	Celgene/BMS
	Study name	ZUMA-1	JULIET	TRANSCEND NHL 001
	Median follow-up (months)	51.1	40.3	18.8
Patients	Median age (range)	58 (23–76)	56 (22–76)	63 (18–86)
	Patients infused	101 (91%)	111 (67%)	269 (78%)
	Lymphomas subtypes	DLBCL/PMBCL/TFL	DLBCL	DLBCL/tiNHL/PMBCL/FL3B
Efficacy	Best ORR, %	83	54	73
	CR, %	58	40	53
	Median PFS (months)	5.9	-	6.8
	PFS, %	39% @ 24 months	31% @ 36 months	44% @ 12 months
	Median OS (months)	25.8	11.1	21.1
	OS, %	44% @ 48 months	36% @ 36 months	58% @ 12 months
Toxicity	Grade ≥ 1 CRS, %	93	58	42
	Grade ≥ 3 CRS, %	13	23	2
	Grade ≥ 1 ICANS, %	64	21	30
	Grade ≥ 3 ICANS, %	28	12	10

CR, complete response; CRS, cytokine release syndrome; DLBCL, diffuse large B-cell lymphoma; FL3B, Follicular lymphoma grade 3B; ICANS, immune effector cell-associated neurotoxicity; PFS, progression free survival; ORR, overall response rate; OS, overall survival; PBMCL, primary mediastinal B-cell lymphoma; tiNHL, Transformed indolent Non-Hodgkin lymphoma.

**Table 3 cancers-13-04274-t003:** Axi-cel (Yescarta^®^): ZUMA-1 vs. Real-world Experience.

CAR T-cells		ZUMA-1	US RWE 1	US RWE 1
Reference	Reference	Jacobson et al. [5]	Nastoupil et al. [10]	Jacobson et al. [9]
Patients	Apheresis	111	298	-
	Patients infused	101 (91%)	275 (92%)	122
	Median age (range)	58 (23–76)	60 (21–83)	62 (21–79)
	≥3 prior lines of treatment (%)	70 (69%)	222 (74.5%)	-
	Eligible for ZUMA-1	100%	43%	38%
	Prior auto-stem cell transplantation	21 (21%)	98 (32.9%)	31 (25%)
	Prior allo-stem cell transplantation	0	7 (2.4%)	4 (3%)
	Bridge therapy	0	158 (53%)	55 (45%)
	ECOG ≥ 2	0	58 (19%)	12 (10%)
Efficacy	Median follow-up (months)	51.1	12.9	10.4
	ORR, %	83	82	70
	CR, %	58	64	50
	Median PFS (months)	5.9	8.3	4.5
	PFS, %	39% @ 24 months	47% @ 1 year	-
	Median OS (months)	25.8	Not reached	Not reached
	OS, %	44% @ 4 years	68% @ 1 year	65% @ 1 year
Toxicity	Grade ≥ 3 CRS, %	13	7	16
	Grade ≥ 3 ICANS, %	32	31	35
	Non-relapse mortality, %	3.7	4.4	-

CR, complete response; CRS, cytokine release syndrome; ICANS, immune effector cell-associated neurotoxicity; ORR, overall response rate; OS, overall survival; PFS, progression-free survival.

**Table 4 cancers-13-04274-t004:** Tisa-cel (Kymriah^®^): JULIET vs. Real-World Experience.

CAR T-cells		JULIET	US RWE 1
Reference	Reference	Jaeger et al. [4]	Pasquini et al. [11]
Patients	Apheresis	115	-
	Patients infused	115	155
	Median age (range)	56 (22–76)	65 (18–89)
	≥3 prior lines of treatment (%)	57 (51%)	-
	Eligible for JULIET	100%	-
	Prior auto-stem cell transplantation	54 (49%)	40 (25.8%)
	Prior allo-stem cell transplantation	-	5 (3.2%)
	ECOG ≥ 1	50 (45)	48 (31%)
Efficacy	Median follow-up (months)	40.3	11.9
	ORR, %	54	61.8
	CR, %	40	39.5
	PFS, %	31% @ 36 months	38.7% @ 6 months
	Median OS (months)	11.1	-
	OS, %	36% @ 36 months	70.7% @ 6 months
Toxicity	Grade ≥ 3 CRS, %	23	11.6
	Grade ≥ 3 ICANS, %	12	7.5

CR, complete response; CRS, cytokine release syndrome; ICANS, immune effector cell-associated neurotoxicity; ORR, overall response rate; OS, overall survival; PFS, progression-free survival.

**Table 5 cancers-13-04274-t005:** Main Bispecific T-cell Engagers.

Bispecific T-cell Engagers	Mosunetuzumab (RG7828)	Glofitamab (RG6026)	Odronextamab (REGN1979)	Epcoritamab
Reference	Schuster et al. [40]	Hutchings et al. [41]	Bannerji R et al. [42]	Clausen et al. [43]
Targets	CD20 × CD3	(CD20)_2_ × CD3	CD20 × CD3	CD20 × CD3
Patients infused	270	171	127	68
Median age (range)	Not reported	68 (44–85)	Not reported	68 (21–84)
Lymphoma subtypes	DLBCL/tFL/FL	DLBCL/tFL/Richter’s transformation/PMBCL	DLBCL/FL/MCL/MZL/other B-NHLs	DLBCL/FL/MCL/others B-NHLs
Median prior lines of therapy (range)	Not reported	3 (1–13)	3 (1–11)	3 (1–18)
Prior CAR T-cells therapy, patients (%)	30 (11.1)	3 (1.8)	29 (22.8)	6 (9)
Median follow-up (months)	6 (since first CR)	13.5 (all)/8.4 (aNHL)/5.8 (iNHL)	3.9	14.1 (all)/10.2 (DLBCL)/15.2 (FL)/10.2 (MCL)
Route of administration	IV	IV	IV	SC
Frequency of administration	D1, D8, D15, every 21 days (until progression)	D1, D8, every 21 days (12 cycles)	D1, D8, every week (12 cycles), then a maintenance every two weeks (until progression)	D1, D28, every 28 days (until progression)
Dose	2.8 mg–40 mg	0.005 mg–30 mg	0.03–320 mg	0.0128–60 mg
OR, %	62.7 (iNHL)/37.1 (aNHL)	71.1 (all)/64.3 (aNHL)/79.2 (iNHL)	92.2 (FL *) /	91 (DLBCL **)/ 80 (FL)/50 (MCL)
CR, %	43.3 (iNHL)/19.4 (aNHL)	63.5 (all)/57.1 (aNHL)/70.8 (iNHL)	75 (FL)	55 (DLBCL)/60 (FL **))/25 (MCL)
Median PFS, months	Not reported	Not reported	Not reported	9.1 (DLBCL)
OS	Not reported	Not reported	Not reported	Not reported
CRS, all grade, %	29	67.3	62.2	59
CRS, grade ≥ 3, %	1.1	3.5	7.1	0
ICANS, all grade, %	4	3.5		6
ICANS, grade ≥ 3, %	1.1	1.2	2.3	3

aNHL, aggressive NHL; CLL, chronic lymphocytic leukemia; CR, complete response; CRS, cytokine release syndrome; DLBCL, diffuse large B cell lymphoma; FL, follicular lymphoma; ICANS, immune effector cell-associated neurotoxicity; Ig, immunoglobulin; iNHL, indolent non-Hodgkin lymphoma; IV, intravenous; MCL, mantle cell lymphoma; MZL, marginal zone lymphoma; ORR, overall response rate; OS, overall survival; PFS, progression-free survival; PMBCL, primary mediastinal B cell lymphoma; SC, subcutaneous; tFL, transformed follicular lymphoma; * dose ≥ 5 mg of odronextamab; ** with highest dose of epcoritamab.

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
