# Peer review of "T-cell Redirecting Therapies for the Treatment of B-cell Lymphomas: Recent Advances"

_cancers, 2021, doi:10.3390/cancers13174274_

Round 1
Reviewer 1 Report
The manuscript by Dr. Messéant and colleagues is a well written review of the available T cell directed therapies for treatment of B cell non Hodgkin lymphomas. The review provides a comprehensive revision of the outcomes of trials of CAR-T cells and bispecific antibodies, with some mention of new and upcoming agents.
Major comments: While I think the authors review all the trials published to date, I found the review somewhat superficial in terms of mechanism of action, mechanism of toxicity, treatment of complications and mechanisms of resistance. There is a significant and growing body of literature discussing these aspects of T cell engaging therapy and would like to see it added.
I think for this paper to be of value for the clinician and investigator, it should include those components.
Minor comments:
- Introduction, paragraph 1. "patients who are refractory or relapse after standard therapies harbor a poor prognosis". This statement is true, but still needs a citation
- (The tables are excellent)
- Page 4, paragraph 3, where is says "half of them received tocilizumab" I assume it means half of all patients.
- Page 5, paragraph 5, replace "lead to the FDA approval of..." to "led to the FDA approval of..."
Author Response
Dear reviewer,
Thank you very much for your observations.
Please see the attachment.
Best regards,

Reviewer 2 Report
This review article is dedicated to the recent advances of the rapidly evolving field of T-cell redirecting therapies. The analysis of the current data regarding CAR-T therapy and bispecific antibodies is highly valuable. However, currently the text of the article would be hardly understandable for readers that are not familiar with the subject. Also the style needs to be improved (abrupt introduction, lack of logical transitions between the sections, excessive use of parentheses in some of the sections).
Minor suggestions:
1) In the introduction section: "B-cell non-Hodgkin lymphomas (B-NHL) benefit from efficient therapies allowing cure or prolonged remission in most patients. However, patients who are refractory or relapse after standard therapies harbor a poor prognosis." Its a general statement which is true for any disease. I think it would be better to describe the current situation with B-NHL. Also the introduction section lacks the rationale and the subject of this review.
2) Section 2.1.2 is called "Confirmed safety and efficacy in DLBCL" but description of safety is limited with one sentence about long-term unexpected toxicities. This mentioning is too brief for most readers, the CRS and ICANS concepts are not introduced. More detailed description of toxicities observed in the studies is needed, or such description should be referenced in text. The whole 2.1 section also lacks evaluation of efficacy of CAR-T in perspective with previous treatment methods.
3) Table 2. TiNHL. The usual abbreviation is tiNHL. PFS should be expressed either in months or years.
4) Table 3. Bridge -> Bridge therapy;
5) Please check the sections of table 3 and 4 for uniformity. For example ≥ 3 prior lines of treatment (%) vs ≥ 3 prior lines (%) ; Prior auto-stem cell transplantation vs Prior autograft, etc.
6) Section name "2.1.2 CAR T-cells earlier in the therapeutic strategy of DLBCL" should be revised as it is not containing predicate, which is essential part of english sentences.
7) "Giving CAR T-cells earlier in the therapeutic strategy may allow production of CAR T-cells from healthier T-cells (with enhanced functionality) after leukapheresis, and treatment of patients with a better performance status and lower tumor burden". This statement should be clarified with necessary references. The term "healthy" regarding cells can be replaced with more suitable scientific description.
Author Response
Dear reviewer,
Thank you very much for your observations.
Please see the attachment.
Best regards
